# Engineering mouse cell fate controller by rational design

Tao Huang[1,2,9], Dong Liu[2,9], Xiaomin Wang[2,9], Junqi Kuang[2,3,4], Manqi Wu[2], Beibei Wang[2], Zechuan Liang[1,2], Yixin Fan[5,6], Bo Chen[2,3], Zhaoyi Ma[1,2], Yu Fu[2], Wenhui Zhang[1,2], Jin Ming[2,3], Yue Qin[2,3], Chengchen Zhao ⓘ[2,4,7] ✉, Bo Wang ⓘ[2,7,8] ✉ & Duanqing Pei ⓘ[2,4] ✉

Cell fate is likely regulated by a common machinery, while components of this machine remain to be identified. Here we report the design and testing of engineered cell fate controller Nanog[BiD], fusing BiD or BRG1 interacting domain of SS18 with *Nanog*. Nanog[BiD] promotes mouse somatic cell reprogramming efficiently in contrast to the ineffective native protein under multiple testing conditions. Mechanistic studies further reveal that it facilitates cell fate transition by recruiting the intended Brg/Brahma-associated factor (BAF) complex to modulate chromatin accessibility and reorganize cell state specific enhancers known to be occupied by canonical *Nanog*, resulting in precocious activation of multiple genes including *Sall4*, *miR-302*, *Dppa5a* and *Sox15* towards pluripotency. Although we have yet to test our approach in other species, our findings suggest that engineered chromatin regulators may provide much needed tools to engineer cell fate in the cells as drugs era.

The emerging paradigm of cells as drugs necessitates strategies to engineer cell-fate in precision. Unlike chemicals, cells are alive and remain responsive to both intrinsic and extrinsic signals, thus presenting formidable challenges not only to engineer but also manufacture under drug-making or GMP conditions. One promising solution may be to engineer cell-fate through approaches exemplified by Yamanaka-factor-based or related methods. The Yamanaka factors, *Oct4*, *Sox2*, *Klf4*, and *Myc*[1], may be considered the first generation of cell-fate regulators. These factors are in fact proto-oncogenes that function to orchestrate complex developmental processes such as germ cell formation, neural development, epithelialization, and metabolism among diverse biological processes in development and pathology reported so far[2-4]. As such, naturally occurring factors such as the Yamanaka factors may not be ideal cell-fate regulators to engineer cell-fate in precision[5].

It is desirable to design a generation of cell-fate regulators based on the following considerations. First, the genome is hardwired with regulatory structures that respond to intrinsic and extrinsic signals. For example, BMP4 mediates the primed to naive transition by reorganizing genome architecture via ZBTB7 family of transcription factors, suggesting that cells are capable of fate transition through a cooperative extrinsic and intrinsic signaling axis[6]. Second, chromatin accessibility dynamics or CAD during cell fate transition may serve as a guide to design factor combinations. The access to DNA elements in the genome is a dynamic characteristic crucial for establishing and maintaining cellular identity. Each cell type has its own unique chromatin accessibility landscape. During reprogramming, whether induced by factor cocktails or chemicals, this landscape transitions from the somatic cell pattern to pluripotency pattern, perhaps relying on a common intrinsic machinery (CiM) for fate conversion[7]. The

[1]College of Life Sciences, Zhejiang University, Hangzhou, China. [2]Laboratory of Cell Fate Control, School of Life Sciences, Westlake University, Hangzhou, China. [3]Institute of Biology, Westlake Institute for Advanced Study, Hangzhou, China. [4]Westlake Laboratory of Life Sciences and Biomedicine, Hangzhou, China. [5]CAS Key Laboratory of Regenerative Biology, South China Institute for Stem Cell Biology and Regenerative Medicine, Guangzhou Institutes of Biomedicine and Health, Chinese Academy of Sciences, Guangzhou, China. [6]University of Chinese Academy of Sciences, Beijing, China. [7]Key Laboratory of Biomedical Intelligent Computing Technology of Zhejiang Province, Hangzhou, China. [8]Zhejiang University of Science and Technology School of Information and Electronic Engineering, Hangzhou, China. [9]These authors contributed equally: Tao Huang, Dong Liu, Xiaomin Wang. ✉e-mail: zhaochengchen@westlake.edu.cn; wangbo@westlake.edu.cn; peiduanqing@westlake.edu.cn

chromatin landscapes of the starting and destination cells can be mapped globally with ATAC-seq and the resulting CAD can inform the designers to attempt to close and open chromatin loci occupied with specific TFs[8–11], which might lead to new combination of factors achieving the same cell fate transitions[12]. Besides, expressing native factors ectopically has intrinsic limitations such as non-specific interaction and aberrant activation of somatic and apoptotic program[13,14]. Rational engineering or directed evolution of reprogramming factors may lead to the discovery of new reprogramming or previously impermissible factors that may improve efficiency and quality of iPSCs[15]. Though several designer transcription factors including fusing transactivation domain to reprogramming factors have been identified[16–19], chimeric reprogramming factors that recruit chromatin remodeling complex to modulate chromatin accessibility remain to be explored. Lastly, proteins have modular structures with well-defined functional domains that can be exploited to generate new functions through engineering. For example, the 12 residue NuRD interacting motif from Sall4 can be engineered to transcription activators to function as transcription repressors[20]. Therefore, it is feasible to engineer cell-fate regulators that may help us control cell fate rationally.

Here we report the design and function of Nanog[BiD] as a representative of engineered cell-fate regulator or eCR. By fusing a 70aa residue BRG1 interacting domain (BiD) of SS18, a factor implicated in naive to somatic transition[21], to the N-terminus of NANOG, we generated a potent eCR that targets chromatin loci enriched with OCT4-SOX2-NANOG motif commonly found in pluripotent stem cells such as iPSCs and ESCs. Mechanistic studies reveal that this eCR facilitates the opening of chromatin loci with fast kinetics and activating genes such as *Sall4, Esrrb, miR-203, Dppa5a* that known to promote pluripotency, but also *Sox15* and *Foxb1* that can enhance the generation of iPSCs. Our results suggest that an engineering approach may yield future cell fate regulators for the cells-as-drugs paradigm.

## Results

### Engineered cell-fate regulator enhances cell reprogramming

It is conceivable that CiM can be engaged and tested with regulators designed and engineered based on CAD and related considerations. For this purpose, based on the concept of CAD, we compared the differences in CAD between mESCs and MEFs to identify chromatin regions that may require opening during cell reprogramming. We then selected candidate factors by referring to binding sites similar to those in the target cell (Supplementary Fig. 1a). In addition to chromatin remodeling factors like *Ss18, Smarca4, Mbd3, Chd7,* and *Rnf2*, we found that pluripotent factors *Nanog, Sox2,* and *Pou5f1 (Oct4)* are critical in maintaining open chromatin accessible for transcription in ESC with high GIGGLE score. Considering their importance as core factors for pluripotent maintenance and cell reprogramming, we explored whether fusion the BiD of SS18, a candidate component of CiM previously identified by CRISPR-based screening in a pluripotent-somatic transition system, with these pluripotent factors could achieve cell reprogramming using the GGSGG linker (Fig. 1a). Besides, many reported TFs that could facilitate iPSC induction (*Klf4, Esrrb, Prdm14, Zfp296*)[22,23] or replace *Oct4* (*Nr5a2, Jdp2*)[24,25] were selected for screening. In addition, *Rax* and *Klf17*, which we found they can enhance somatic cell reprogramming (not published results) were also selected. We initially attempted cell reprogramming using a single BiD factor, but only a limited number of clones were produced by Oct4[BiD]. Therefore, we hypothesize that introducing additional reprogramming factors alongside the BiD factors that may facilitate reprogramming. Subsequently, we tested these BiD factors in combination with 11 wild-type TFs to convert MEFs to iPSCs and demonstrate that Oct4[BiD] with *DsRed, Esrrb,* or *Nanog*, as well as Nanog[BiD] with *Jdp2* or *Oct4*, could generate iPSCs with varying efficiencies (Fig. 1b, c). As the most efficient combination merged from these attempts is the combination of

Nanog[BiD]+*Oct4*, followed by Oct4[BiD]+*Esrrb* (Fig. 1c), we decided to focus on the front runner−Nanog[BiD]+*Oct4*. We further optimized the process by varying the length of culture in iCD3 or 2iL (Fig. 1d). Using the best condition, *Nanog+Oct4* (hereafter referred as Nanog[WT]) could hardly induce iPSCs, while more than 600 iPSCs colonies can be induced by *Oct4*+Nanog[BiD] (hereafter referred to Nanog[BiD]) from 15,000 MEFs (Fig. 1e, f). The reprogramming efficiency is approaching to that of OKS which we have shown previously to be superior to the original OKSM[26], the gold standard combination for iPSC generation. Interestingly, when we compare BiD with full-length or IDR of SS18, BiD is the most effective (Supplementary Fig. 1b). As expected, our findings demonstrate that Nanog[BiD] iPSCs exhibit transcriptional similarities to mESCs, also can generate chimera with blastocysts injection (Supplementary Fig. 1c, d, f). Moreover, we have collected RNA-seq data from three additional cell reprogramming strategies conducted by other research groups: seven factors-derived reprogramming (GSE127927)[25], OKS derived reprogramming (GSE93029)[27], and small-molecule induced reprogramming (GSE48252)[28]. As shown in Supplementary Fig. 1e, we demonstrate that NanogBiD iPSCs exhibit transcriptional similarities to OKS-iPSCs and other types of iPSCs, particularly the small-molecule-induced ones. Genomic PCR detection revealed that all the iPSC is generated by wild type or synthetic *Oct4* and *Nanog*, but not other reprogramming factors (Fig. 1g). Moreover, we isolated mouse tail tip fibroblasts (TTFs) and mouse neonatal fibroblasts (MNFs) from OG2 transgene mice, to investigate the potential of our system in reprogramming other cell types. Our results demonstrate that all these types of cells could be efficiently reprogrammed to Oct4-GFP positive iPSC colonies that can be stably passaged with key pluripotency features (Supplementary Fig. 1g–i), indicating the validity and generalizability of our system.

SS18 is a component of BAF complex whose components have been shown to enhance reprogramming[29]. We then compare BAF subunits with Nanog[BiD] and show that even the best subunit *Smarca4* among them is 40× less efficient (Supplementary Fig. 1j), suggesting that eCR engineering offers a more direct approach to engage the candidate CiM.

### Direct engagement of BAF complex by Nanog[BiD]

Native NANOG interacts with ~130 proteins including TFs, chromatin-modifying complexes, and basal transcriptional machinery members through classic protein-protein interactions or PPIs[30]. By fusing BiD to *Nanog*, we expect a more direct interaction with BAF. To this end, we collected MEF cells undergoing reprogramming with wild type or synthetic *Nanog* in combination with *Oct4* for 1 day and performed IP-MS experiments (Fig. 2a). Candidate Nanog[BiD] partners were selected based on detection in at least two independent MS experiments. Initially, we employed Log2 > 1.2 and *p*-value < 0.05 as the criteria to determine the presence of positive proteins detected by IP-MS. Only positive proteins presented in both replicates of the same treatments would be considered as candidates for the next differential analysis between Nanog[BiD] and Nanog[WT]. This analysis eventually identified 19 high-confidence interacting partners for Nanog[BiD] and 2 for Nanog[WT], and 531 shared by both. Among the 19 partners for Nanog[BiD], 9 are BAF subunits, SMARCA4/BRG1, SMARCC1/2, SMARCD1/3, SMARCE1, ACTL6A, ARID1A/1B (Fig. 2b). However, we did not detect PBAF or ncBAF specific subunits such as BRD7/9, PHF10, GLTSCR1, or PBRM1 as expected as SS18 is not involved in PBAF complex assembly. Those results suggest that Nanog[BiD] directly engages cBAF to facilitate chromatin opening. The interaction between Nanog[BiD] and BAF subunits was further confirmed by Co-IP experiments (Fig. 2c). Recent report suggests that the interaction between SS18 and BRG1 is largely disrupted when residues A65, L54, and I32 are mutated to glutamic acid, referred as 3M[31]. We then tested this by mutating them in Nanog[BiD] and showed that the resulting mutant became ineffective in reprogramming (Fig. 2d, e; Supplementary Fig. 2a, b). These results suggest that

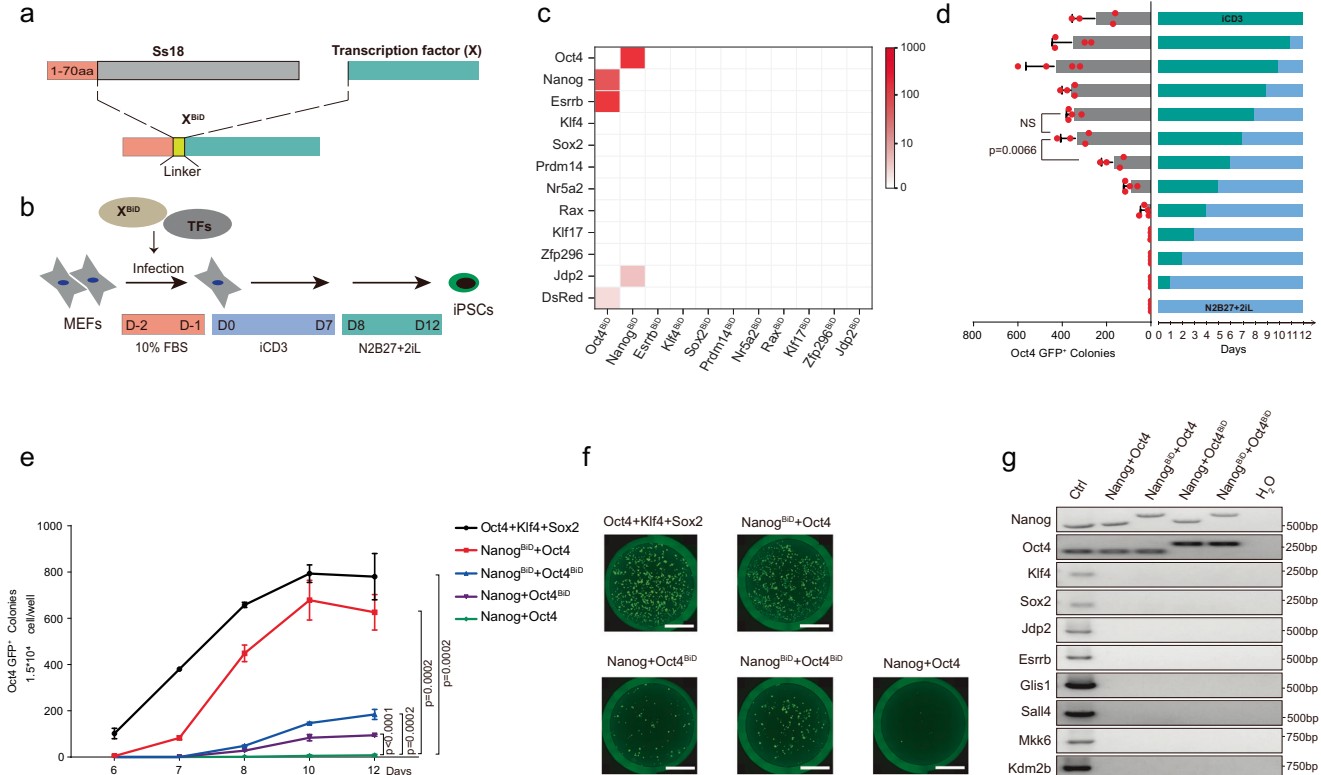

**Fig. 1 | Nanog^BiD facilitates somatic cell reprogramming. a** Architecture of the synthetic factors. The N-terminus 70aa domain from SS18 was fused to the N-terminus of full-length transcription factors. **b** Schematic diagram illustrating the process of synthetic BiD factors induced cell reprogramming. **c** Heatmap of numbers for Oct4-GFP⁺ colonies of MEF cells reprogramming by different combinations. Red rectangles represent mean data for $n = 3$ biological replicates. **d** Number of iPSC colonies induced from MEFs infected by Nanog^BiD+Oct4 under different medium switch strategies. Bars are mean with SD, and red plots are individual data points for $n = 4$ biological replicates. Statistical significance relevant to number of Oct4-GFP⁺ colonies at Day 7 was measured with two-tailed unpaired *t*-test. *P* values were performed on graph. NS, nonsignificant. **e** Number of Oct4-GFP⁺ colonies during the process of *Oct4+Sox2+Klf4*, Nanog^BiD+*Oct4*, *Nanog*+Oct4^BiD, Nanog^BiD+Oct4^BiD, and *Nanog+Oct4* mediated reprogramming. Bars are mean ± SD for $n = 3$ biological replicates. Statistical significance relative to colonies number of *Nanog+Oct4* at Day 12 was measured with two-tailed unpaired *t*-test. *P* values were performed on graph. **f** Representative images of different reprogramming strategies induced colonies on day 12. Scale bar, 5 mm. **g** Detection of plasmid integration by PCR. Gels shown are representative of $n = 2$ independent experiments. Source data are provided as a Source Data file.

the directed engagement of cBAF by Nanog^BiD is critical for cell fate conversion in iPSC generation.

## Nanog^BiD-BAF complex accelerates chromatin opening

A previous study has documented that the OKS-*Brg1-Baf155* combination yielded 12 times more induced pluripotent stem cells (iPSCs) compared to the OKS combination. This was attributed to the OKS-*Brg1-Baf155* combination's ability to induce a euchromatic chromatin state and facilitate the binding of reprogramming factors to key gene promoters[23]. Given the observed physical association between the BAF complex and Nanog^BiD, we then investigated whether cBAF expedites Nanog^BiD reprogramming by deconstructing compact somatic chromatin and establishing an open chromatin state that is conducive to pluripotency. To this end, we initially analyzed the transcriptional changes in Nanog^BiD and Nanog^WT reprogramming by harvesting cells at various time points (day 0, day 1, day 3, day 5, day 7, day 8, day 10, and day 12), along with MEFs and mESCs, as control, for RNA-sequencing. Based on the transcriptional dynamics, all the differentially expressed genes can be categorized into 12 different clusters (Fig. 3a). Notably, three patterns emerge to depict significant dynamics during the reprogramming process and obvious difference between Nanog^BiD and Nanog^WT (Fig. 3b). Principal component analysis shows that there is accelerated transition from somatic state to pluripotent state with Nanog^BiD compared to Nanog^WT (Supplementary Fig. 3a). Particularly, on day 5 of the reprogramming process, discernible

divergence in gene expression between Nanog^BiD and Nanog^WT emerged, subsequently exhibiting a progressive expansion. Similar correlations can be observed among specific genes, such as pluripotent genes *Sall4* and *Sox2*, or the somatic specific genes *Dab2* and *Thbd* (Supplementary Fig. 3b). As such, we picked day 5 as the time point for CUT&Tag experiment (Supplementary Fig. 3k). Gene ontology analysis of the genes associated with the cluster 1 revealed genes activated by Nanog^BiD are associated with stem cell differentiation, embryonic development, and pattern specification, whereas the genes associated with the cluster 12 revealed genes repressed by Nanog^BiD are related to extracellular structure organization, and cellular response to growth factor stimulus (Fig. 3c, d).

Given the clear role of BAF complex in reprogramming, we wish to resolve how this complex regulates chromatin dynamics in Nanog^BiD reprogramming by ATAC-seq (Fig. 3e). All the performed samples were collected as same time points as RNA-seq and the close to open (CO) and open to close (OC) peaks were divided into 9 subgroups (Supplementary Fig. 3c). Counting peaks further shows that the number of peaks from OC1-9 has not significant difference, but the number of total peaks from CO1-CO8 in Nanog^BiD (5674) are two times more than Nanog^WT (2735), suggesting that BAF complex may regulate CO but not OC. Besides, more than 80,000 peaks (CO9) remain at closed state at day 12 but open in ESCs. Nearly 40,000 peaks (OC1) are open in MEFs and closed in day 0 (Supplementary Fig. 3d). Later analysis demonstrates that BAF complex regulates chromatin accessibility at early

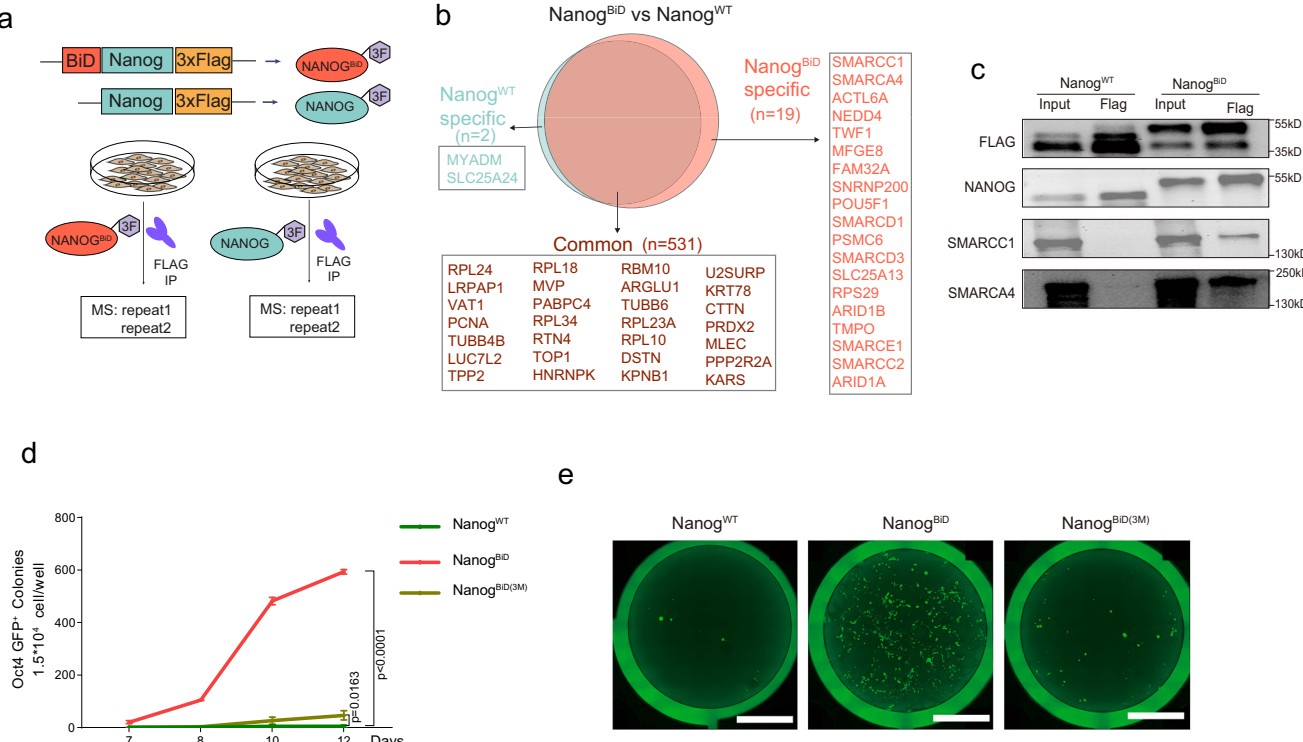

**Fig. 2 | NanogBiD interacts with the BAF complex to enhance reprogramming efficiency. a** Proteins of reprogramming samples at day 1 performed in two replicates for FLAG pull-down followed by MS analysis. **b** Venn diagram shows numbers of protein common and specific identified in NanogWT or NanogBiD associated reprogramming samples. Proteins are listed as official gene symbols. **c** Flag Co-IP followed by western blot analysis of SMARCC1, SMARCA4, FLAG, and NANOG. Blots shown are representative of $n = 2$ independent experiments. **d** Percentage of homogeneous Oct4-GFP positive colonies induced in NanogBiD, NanogBiD(3M), and NanogWT mediated reprogramming at different time points. Data are presented as mean ± SD for $n = 3$ biological replicates. Statistical significance relative to colonies number of NanogWT at Day 12 was measured with two-tailed unpaired $t$-test and precise $p$ values are shown on graph. **e** Representative images of different reprogramming strategies induced colonies on day 12. Scale bar, 5 mm. Source data are provided as a Source Data file.

stage and neither CO9 nor OC1 are not the key chromatin loci that determine the successful reprogramming. If these two subgroups were to be included in the heatmap, additional subgroups would be over-compressed and not visible, thus, not included (Fig. 3e).

To gain mechanistic insight into chromatin dynamics, we perform motif analysis for TFs associated with OC and CO peaks (Fig. 3e, Supplementary Fig. 3e). We show that OC loci are enriched with somatic TFs such as TEAD, ATF, and AP-1 family TFs in both systems as expected. However, motifs for SOX and RFX family TFs, such as SOX15, SOX2, SOX17, and RFX2 are enriched in CO peaks at early stage in NanogBiD but not NanogWT (Fig. 3f). These results suggest that NanogBiD recruit BAF complex to open pluripotent chromatin loci. To further identify the differences, we focus on peaks from OC2-9 and CO1-8 in both NanogBiD and NanogWT and calculated those loci that failed to close and failed to open (Fig. 3g). More than 30% of early CO peaks are sensitive to NanogBiD but failed to open in NanogWT. Importantly, less than 10% of early OC is different between two systems, suggesting that the primary role of NanogBiD−BAF complex is for chromatin opening.

As expected, NanogBiD-sensitive CO peaks are dominated by motifs for TFs such as SOXs, and OC peaks enriched for somatic TFs of BACH1/2, FOS, FRA, and ATFs (Fig. 3h, i, Supplementary Fig. 3f). Consistently, CO and OC peaks are associated with loci for mESC and MEF specific enhancers, respectively (Supplementary Fig. 3g). At the chromatin accessibility dynamics level, NanogBiD also exhibits quicker and more similar to pluripotency than NanogWT (Supplementary Fig. 3h). To validate above analysis, we expressed SOX family TFs with *Nanog+Oct4* for reprogramming and show that *Sox15*, *Sox3*, and *Sox1* significantly promote iPSCs generation (Supplementary Fig. 3i). Consistent with motif enrichment results, gene ontology of NanogBiD-

sensitive CO1-CO5 show that NanogBiD appears to divert cell fate towards to stem cell such as leukemia inhibitory factor (LIF) activation, G1/S transition, and stem cell maturation (Fig. 3j). Among these GO terms, miR-302/367 family responds to LIF signaling and fails to open in NanogWT, but promotes iPSCs generation with *Nanog+Oct4* as reported (Fig. 3k, Supplementary Fig. 3j). Furthermore, integrating analysis with RNA-seq, we identify 211 genes between cluster 1 and the NanogBiD-sensitive CO regions (Fig. 3l). As expected, these genes are related to maintenance of pluripotent stem cell (Fig. 3m). Among them, we can identify 43 TFs, and show that at least *Sox15*, *Foxb1*, *Sall4*, *Klf4*, *Sox2* and *Olig3*, can facilitate *Nanog+Oct4* reprogramming (Supplementary Fig. 3j). Of note, we performed CUT&Tag for Flag-tagged NanogBiD and NanogWT reprogramming at day 5 using anti-FLAG/BRG1/H3K27ac antibodies and showed that they are more significantly enriched in NanogBiD than NanogWT (Supplementary Fig. 3k). Together, these results suggest that NanogBiD accelerates opening of pluripotent specific chromatin loci.

## NanogBiD targets BRG1 to pluripotent loci

We then wish to probe the mechanism through which NanogBiD promotes chromatin opening by performing CUT&Tag experiments on day 5 reprogramming MEFs for NanogBiD and NanogWT, BRG1, H3K27ac, and H3K4me1. Quantification of Flag-tagged NANOG signal indicates that NanogBiD overlaps significantly with NanogWT (Fig. 4a), suggesting that grafting BiD onto *Nanog* does not change DNA binding specificity of NANOG. In contrast, we show that BRG1 binding sites are doubled in NanogBiD compared to NanogWT, validating the utility of this engineered factor to open chromatin loci more efficiently (Fig. 4b). We also show higher correlation and more common peaks in NanogBiD than

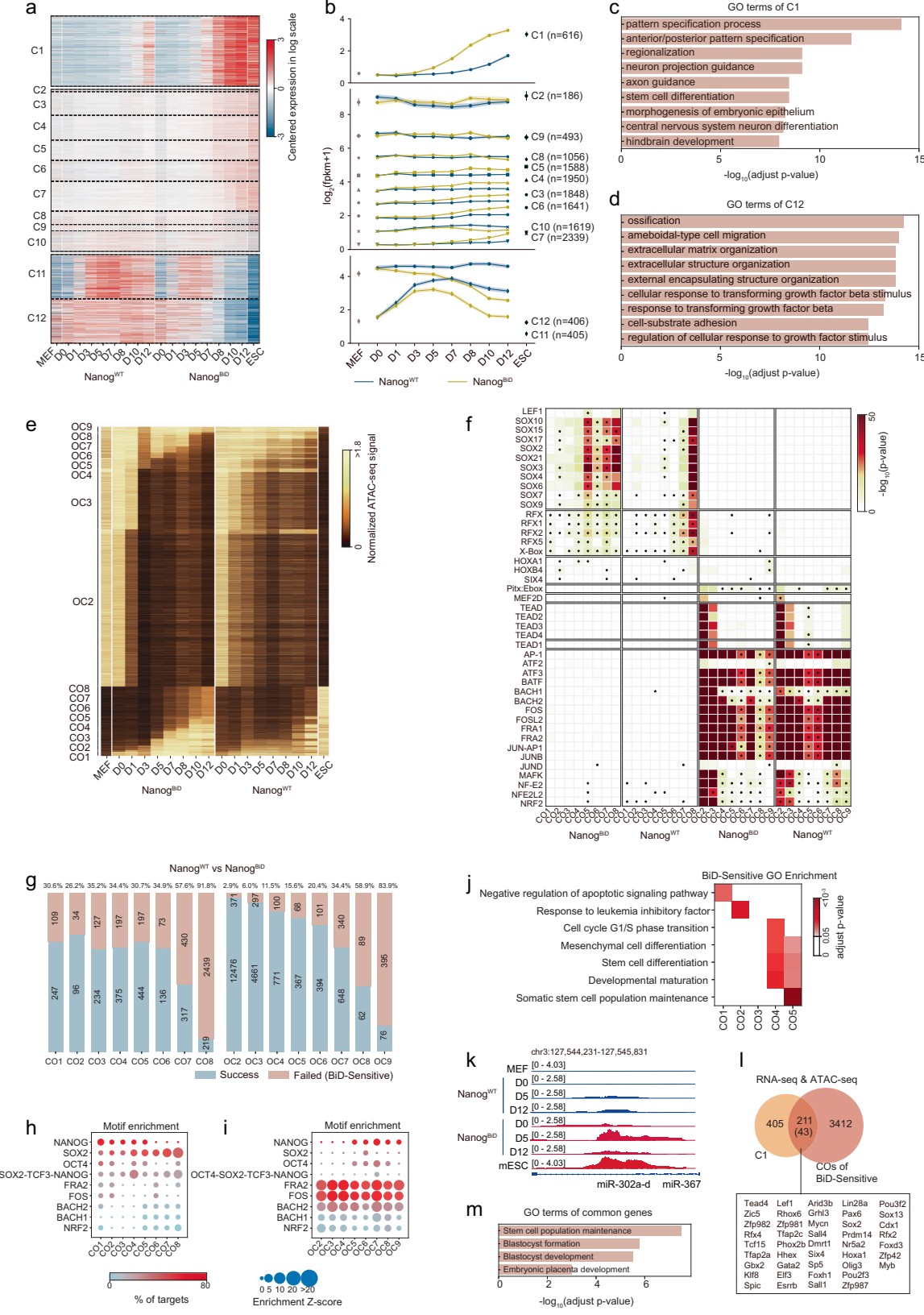

that of Nanog[WT] (Supplementary Fig. 4a, b). These results reveal that Nanog[BiD] targets BRG1 to open closed chromatin.

We further classified CUT&Tag peaks into three groups: BiD-specific, WT-specific, and common (Fig. 4c). Among them, BRG1 exhibits distinct distribution with ~35% regions (10,376) occupied by NANOG and BRG1 simultaneously (refer as 0101), ~57% (17,316) by both in a

much weaker manner (refer as 0100). We also observed an interesting pattern for BRG1 vs Nanog[WT] and Nanog[BiD] in the common group, with 14,212 (refer as 1101) regions occupied by BRG1 strongly with NANOG, again suggesting that Nanog[BiD] also targets BRG1 to those regions. There are also 6741 regions (refer as 1100) with relatively weak BRG1 binding with Nanog[BiD] in the common group. Together, these results

**Fig. 3 | Nanog^BiD increases chromatin accessibility and pluripotent gene expression. a** Heatmap of the expression dynamics during Nanog^WT and Nanog^BiD reprogramming path as well as MEF and mESC for all differentially expressed genes. These genes were grouped into 12 clusters according to their similarity of expression dynamics. D indicates day. **b** Line plots show the expression dynamics trends of different gene clusters defined in (**a**). **c, d** Functional enrichment analysis of the C1 and C12 genes, respectively. **e** Heatmap of the chromatin accessibility dynamics during Nanog^WT and Nanog^BiD reprogramming path as well as MEF and mESC. Loci of open chromatin were arranged into groups depending upon the day of Nanog^BiD reprogramming they changed from closed to open (CO) or open to closed (OC). **f** Motif of transcription factors significantly enriched in each group of chromatin loci of Nanog^BiD reprogramming path (as defined in (**e**)) and Nanog^WT

reprogramming path (as defined in Fig. S3e.). Motifs with at least 3-fold enrichment and less than $1 \times 10^{-5}$ p-value are marked with asterisk. **g** Bar plots show the number of peaks specifically affected by Nanog^BiD (Failed) and both affected by Nanog^BiD and Nanog^WT. **h, i** The motif enrichments of pluripotency factors and somatic factors in the Nanog^BiD specifically affected CO and OC regions, respectively. The dots were colored by the percentage of each motif in targets. **j** Functional enrichment analysis of Nanog^BiD specifically affected CO regions demonstrated in heatmap. **k** Genome track view of the ATAC-seq data for the pluripotency gene cluster miR302a-d. **l** Venn diagram shows the overlapped gene number of C1 and Nanog^BiD specifically affected CO genes. The 43 transcription factors are listed. **m** Functional enrichment analysis of the overlapped genes in (**l**).

appear to suggest that Nanog^BiD targets BRG1 to a much wider portion of the genome compared to Nanog^WT (Fig. 4d). Both BRG1 and NANOG binding changes tend to be at distal regions, i.e., mainly enhancer remodeling (Fig. 4e). Consistently, we show a similar result when we analyzed NANOG binding and H3K27ac changes (Supplementary Fig. 4e). Furthermore, NANOG motif searching from FLAG peaks revealed Nanog^BiD binding regions contain more ESC-specific NANOG motifs (Supplementary Fig. 4c, d). These results suggest that Nanog^BiD and BRG1 have similar genomic binding patterns.

To assess the impact of Nanog^BiD and BRG1 co-occupancy described above, we analyzed the ATAC-seq signals in defined regions. As shown in Fig. 4f, Nanog^BiD occupancy correlates positively with greater chromatin accessibility in the 0101 regions. A similar relationship could also be found in the 0100 regions (Supplementary Fig. 4f). Consistent with chromatin accessibility, H3K27ac and H3K4me1, both marking enhancers, are enriched more in 0101 and 0100 areas in Nanog^BiD than that of Nanog^WT (Fig. 4g, h; Supplementary Fig. 4g, h). In contrast, there is no such positive correlation between ATAC-seq and Nanog^BiD or Nanog^WT in the common group from day 3 to day 12, but an evidently negative one (Supplementary Fig. 4i, l). Similar relationships are observed for H3K27ac and H3K4me1 modifications in similar regions (Supplementary Fig. 4j, k, m, n). For these four regions, we extracted chromatin state annotations for enhancer and promoter and showed that both NANOG and BRG1 modulate epigenetic modification in enhancer, but not promoter regions (Fig. 4i). Of note, both 0101 and 1101 have higher enrichment scores than their counterpart 0100 and 1100, suggesting that BRG1 occupancy promotes enhancer activation.

To investigate the relationship between BRG1 occupancy and chromatin accessibility and histone modification, we analyzed BRG1 binding at regions with different ATAC-seq signals and histone modifications. First, the day 5 ATAC-seq samples were calculated and Nanog^BiD and Nanog^WT specific regions were defined. Then, we computed the numbers of overlap regions between ATAC-seq and CUT&Tag. The percentage of overlap regions is calculated by counting overlap peaks in CUT&Tag regions (Fig. 4j). The distinct ratio between 0101 and 0100 is higher in Nanog^BiD−specific ATAC-seq regions than that of Nanog^WT−specific regions indicating that BRG1 binding increases chromatin accessibility. Then, Nanog^BiD and Nanog^WT−specific enhancers were analyzed with H3K27ac and H3K4me1 modification to show that the levels of BRG1 and NANOG binding correlate well with epigenetic modifications at active enhancers (Fig. 4k, l).

The genome-wide correlation analyses described above can also be validated at specific gene loci, revealing specific genes targeted by Nanog^BiD and BRG1. As shown in Fig. 4m, genes such as *Sall4*, *Esrrb*, *miR-302*, *Dppa5a* are known to promote iPSC formation[32–34], validating the positive role of the Nanog^BiD and BRG1 interaction. We then tested select candidates and showed that they indeed promote iPSC generation (Fig. 4n, o; Supplementary Fig. 3i, j), including those not previously known such as *Sox15*, *Foxb1*, and *Olig3*. It is likely that the sum of these downstream targets may account for the efficiency boost from the engineered factor Nanog^BiD.

## Discussion

In this report, we hypothesize that cell fate is intrinsically controlled by a common machine or cell-intrinsic machine CiM. This hypothesis is consistent with our current knowledge about development, for example, the precision in generating all cells in our body from a fertilized egg, or the same precision achieved in reverse for converting somatic cells to pluripotency in vitro with diverse methods. The components of CiM remain largely unknown, although many candidates have been revealed in previous studies, especially those related to the epigenetic machinery. Here we took an engineering approach, inspired by the "build to understand" principle, to probe a candidate component of CiM- the BAF complex. Specifically, we designed and tested an engineered cell-fate regulator or eCR such as Nanog^BiD, and showed that it can affect open pluripotent chromatin and generate iPSCs with greater efficiency.

Notably, the engineered reprogramming factors cooperatively bind DNA and regulate chromatin accessibility in a synergistic manner. Subsequent mechanistic study uncovers genes (*Sox15* and *Foxb1*) and their combinations, suggesting that this approach may be generalized to design eCRs not only for basic research in stem cell biology but also for therapeutic applications. As our understanding of cell fate control and genome architecture progresses rapidly, this approach may become more fruitful and efficient. While we only leveraged our knowledge on *Brg1*/BAF and *Nanog* for designing Nanog^BiD in the context of somatic to pluripotent cell fate transition (Fig. 4p), it would be interesting to consider the other chromatin remodeling complexes such as INO80, ISWI, and CHDs with much broader applications such as the generation of other cell types, for example, NK cells. We have yet to test our approach in other species, including human cells. To adopt our engineering approach to human cell reprogramming, one needs to consider the multifaceted and unique nature of human cell reprogramming. It is also essential to map the CADs of human cells and design corresponding factors in a similar fashion as described here.

While we focused on one eCR, our designed panel has yielded additional combinations which may be investigated further in the near future. Furthermore, the rest of candidates such as Klf4^BiD, Sox2^BiD, Nr5a2^BiD, Klf17^BiD, and Zfp296^BiD (Fig. 1c) may be active in other cell fate control systems or under conditions not tested here. Further studies would be required to demonstrate their potential as eCRs. While our data suggests that DNA binding specificity remains unaffected by grafting BiD onto *Nanog*, it remains unknown if other properties have been altered. For instance, recent studies have reported that activation domains (ADs) can increase TFs residence time on chromatin binding sites and recruit co-factors to achieve transcription activation[35]. Others have reported that ADs control the interaction TFs with coactivators by driving phase separation[36]. Therefore, additional considerations such as the structural location of both DNA binding domains (DBDs) and ADs within proteins should be evaluated for better designs. Nevertheless, our studies described here may encourage similar efforts to optimize the design of more precise and universal eCRs.

Our results may also help resolve an earlier puzzle that *Nanog* was not part of the original Yamanaka factors, *Oct4-Sox2-Klf4-Myc*, despite

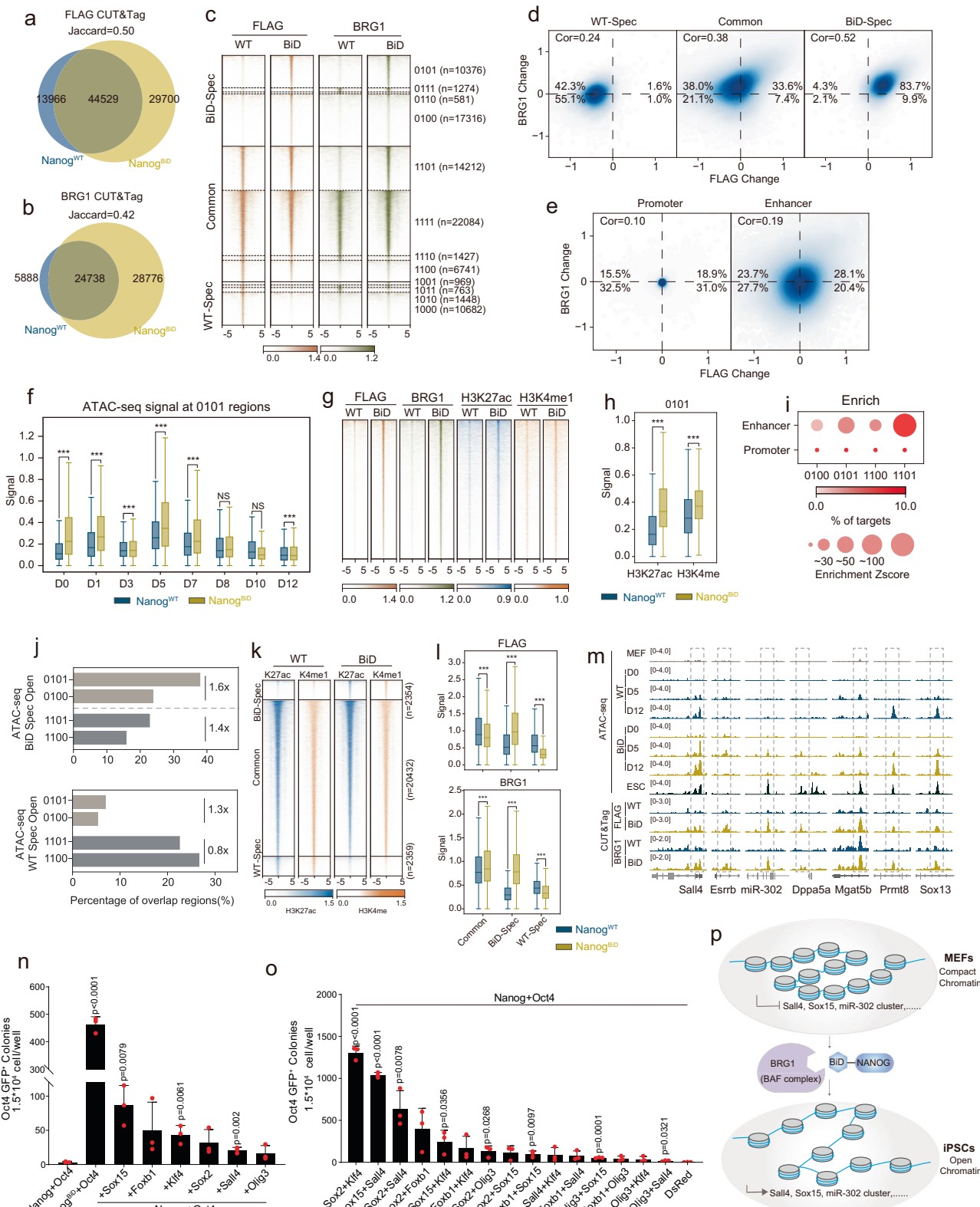

the fact *Nanog* was shown to the gateway to naive pluripotency[1,37]. Our results suggest that Nanog[BiD] is effectively replacing *Sox2*, *Klf4*, and *Myc* functionally. By directly accessing BAF, the eCR is switching on part of the CiM as effectively as the combined power of *Sox2-Klf4*. Further studies may reveal the mechanistic differences between eCR and native factors *Sox2* and *Klf4*. Since native factors such as *Sox2* and *Klf4*

were evolved to control developmental processes, not intended to induce pluripotency, it is likely only a small fraction of their regulatory activities or combinations of are contributing to pluripotency induction. Indeed, in one extreme case, *Myc*, as part of the original Yamanaka factor combination, becomes deleterious in pluripotency induction when Vitamin C is present in the culture media[26]. This

**Fig. 4 | Nanog[BiD] recruits the BAF complex to open pluripotent chromatin.**
**a**, **b** Venn diagrams show the specific and overlapped NANOG or BRG1 CUT&Tag peak numbers at D5 of the Nanog[WT] and Nanog[BiD]. **c** Heatmap of the NANOG and BRG1 binding loci at D5 in Nanog[WT] and Nanog[BiD]. **d**, **e** Scatter plot shows the relationship of NANOG or BRG1 binding change in the specific and overlapped regions defined in (**a**) promoter regions and enhancer regions, respectively. The Spearman correlation coefficients were calculated for each type of region. **f** Boxplot of the ATAC-seq signal at each stage of two reprogramming paths in 0101 regions. **g** Heatmap of the NANOG, BRG1, H3K27ac, and H3K4me1 signal at D5 of two reprogramming paths in 0101 regions. **h** Boxplot of the H3K27ac and H3K4me1 signal at D5 of two reprogramming paths in 0101 regions. **i** Enrichment analysis of four classes of regions (0100, 0101, 1100, 1101) in the enhancers and promoters. The dots were colored by the percentage of each motif in targets. **j** Percentage of specific open loci in the above-mentioned four classes of regions at D3, D5, and D7 during reprogramming. **k** Heatmap of enhancers regions defined by H3K27ac and H3K4me1 in Nanog[WT] and Nanog[BiD]. **l** The NANOG and BRG1 signal in the specific and common regions defined in (**k**). The box plots (**f, h, j, l**) indicate the medians (centerlines), first and third quartiles (bounds of boxes), and 1.5 multiplied by interquartile range (whiskers). Statistical analysis was performed using student's two-sided *t*-test. ***$p < 0.0001$, NS nonsignificant. **m** Genome track view of the ATAC-seq and CUT&Tag data for the C1 genes which have NANOG and BRG1 Co-binding and CO (closed to open) pattern affected by Nanog[BiD]. **n, o** The effects of *Sox15, Foxb1, Klf4, Sox2, Sall4*, and *Olig3*, or different combinations in *Nanog+Oct4* mediated reprogramming system. Data are presented as mean ± SD (*n* = 3 biological replicates), *p* values are determined by two-tailed unpaired *t*-test. **p** A model for fibroblasts reprogramming with BiD factors. Precise *p* values were provided in the Source Data file. WT indicates Nanog[WT]. BiD indicates Nanog[BiD].

evidence supports an engineering approach to cell fate control by rationally removing unnecessary or deleterious elements from native factors.

# Methods

## Animals
Oct4-GFP (OG2) reporter-allele-carrying mice (CBA/CaJ x C57BL/6J, 7–8 weeks, male and female) were obtained from The Jackson Laboratory. The wild-type female 129 mice (129S2/SvPasCrl, 7–8 weeks) and wild-type ICR mice (ICRCF1/Crl, 7–8 weeks, male and female) were purchased from Vital River Laboratory Animal Technology Co., Ltd (Beijing). All animals were housed in a temperature-controlled room with a 12 h light/dark cycle, 20–26 °C ambient temperature, and 40–70% humidity. Diet and water were available ad libitum. The animal studies were performed according to the applicable guidelines and regulations of the Institutional Animal Care and Use Committee of Westlake University (Animal Protocol No. 23-107-PDQ), Hangzhou, China.

## Cell lines
OG2 MEFs were isolated from E13.5 embryos (female or male) by crossing male OG2 mice to 129 female mice. In briefly, the embryos were isolated and cut into small pieces. Then the tissues were digested with digestive solution (0.25% trypsin: 0.05% trypsin = 1:1; GIBCO, 25200072, 25300054) for 15 min at 37 °C. The isolated cells were plated onto 0.1% gelatin (Millipore, ES-006-B) coated culture dish. Mouse neonatal fibroblasts (MNFs) were isolated from the dorsal skin dermis of day 2–3 neonatal pups (female or male). Mouse TTFs were isolated from 8-week-old mice (female or male). Skin tissues or adult tails were sliced into small pieces and plated onto 6 cm culture dishes coated with 0.1% gelatin solution. Typically, fibroblasts migrate outwards from the tissues and reach up to approximately 90% confluence after 1 week. Plat-E cells were obtained from Guangzhou Institutes of Biomedicine and Healthy, Chinese Academy of Sciences. All cells and tissues were cultured in DMEM/high-glucose (Hyclone, SH30022-2B) supplemented with 10% FBS (Natocor, NTC, SFBE), 1% GlutaMax (GIBCO, 35050061), 1% nonessential amino acids (NEAA, GIBCO, 11140035), and 1% sodium pyruvate (GIBCO,11360070) at 37 °C in an atmosphere of 5% $CO_2$.

Mouse ESCs derived from embryos (Male or female) of OG2 mice and iPSCs (Male or female) derived from this study were maintained feeders-free with N2B27 + 2iL medium (50% (v/v): DMEM/high-glucose (Hyclone), 50% (v/v) knock out DMEM (GIBCO, 10829018), N2 (GIBCO, A1370701), B27 (GIBCO, 17504044), 1% sodium pyruvate (GIBCO), 1% NEAA (GIBCO), 1% GlutaMAX (GIBCO), 0.1 mM beta-mercaptoethanol (GIBCO, 21985023), 1000 U/ml LIF (Millipore, ESGE107), 3 mM CHIR99021 (Sigma, SML1046), and 1 mM PD0325901 (Sigma, PZ0162)). All cell lines were tested negative for mycoplasma with the Kit from Lonza (LT07-318).

## Plasmids and molecular cloning
pMXs retroviral vectors expression mouse *Sox2, Klf4*, and *Oct4* were purchased from Addgene. Full-length coding sequences of individual factors were amplified and cloned into the pMXs backbone vector.

## iPSCs generation
Plat-E cells were transfected with plasmids using polyethylenimine reagent (PEI, Yeasen, MW40000) to produce retroviral supernatants. OG2-MEFs were plated onto 24-well plate at $1.5 \times 10^4$ cell density per well, and then infected with the retroviral supernatants. After two rounds of 24 h infection, which we designated as day 0, the medium was changed to iCD3 reprogramming induing medium[20]. The fresh medium was changed daily. After seven days or at designed time point, the medium was switched to N2B27 + 2iL medium. Around day 12, Oct4-GFP positive colonies were scanned by living cells station (Keyence, Japan), and counted using Image-J software (v1.54f, NIH).

## RNA-seq
Total RNA was extracted from cells using FreeZol reagent (Vazyme, R711-01) following the manufacturer's instructions. For RNA-seq, about 1 μg of total RNA was used for sequencing library construction by the VAHTS Universal V10 RNA-seq Library Prep Kit for Illumina (Vazyme, NR606) and was sequenced on an Illumina Novaseq PE150 platform. The RNA-seq reads were trimmed using Trim Galore (v0.6.4)[38] and then mapped to the mm10 reference genome with HISAT2 (v2.2.1)[39]. StringTie (v2.2.1)[40] was used to quantify the transcription level of each gene in each sample into FPKM (Fragments per kilobase of exon model per million mapped reads). GFOLD (v1.1.4)[41] was used to perform differential expression analysis between conditions. The differentially expressed genes were identified with gfold value > 0.5.

## Co-immunoprecipitation and western blot
For co-immunoprecipitation, cells were lysed with NP-40 buffer (50 mM Tris pH 7.4, 200 mM NaCl, 10% Glycerol, 1% NP40, 1 mM EDTA) for 2 h. Whole cell lysates were incubated overnight at 4 °C with anti-FLAG M2 magnetic beads (30 μl, Sigma, M8823) for 3 × FLAG-tagged Nanog[WT] or Nanog[BiD]. Beads were then washed with cell wash buffer (50 mM Tris pH 7.4, 200 mM NaCl, 10% Glycerol, 0.01% NP40, 1 mM EDTA) for three times, and eluted with loading buffer (4% SDS,10% 2-Mercaptoethanol, 20% Glycerol, 0.004% Bromophenol blue, 0.125 M Tris pH 6.8) for 10 min at 95 °C. For western blotting, total protein or IP eluates were separated by SDS-PAGE and transferred onto the PVDF membrane (Millipore). After being blocked with 10% nonfat milk for 2 h at room temperature, the membranes were sequentially incubated with primary and secondary antibodies. Finally, the bands were detected using an enhanced chemiluminescence kit (Thermo). Primary antibodies used were: anti-NANOG (Abcam, ab214549, 1:1000), anti-FLAG (Sigma, F1804, 1:1000), anti-SMARCA4 (Abcam, ab110641, 1:5000), anti-SMARCC1 (Abcam, ab305037, 1:1000).

## Mass spectrometry analysis

IP samples were subjected to fractionation via SDS-PAGE and subsequently digested with trypsin. Fractions were then separated using Acclaim™ PepMap™ 100 C18 column (Thermo, 164941) over 140 min (100 min of 2–22%, 20 min 22–28%, and 12 min of 28–36% gradient of B buffer (80% acetonitrile and 0.1% formic acid in $H_2O$)), following with two steps washes: 2 min of 36–100% and 6 min of 100% B buffer with an Easy-nLC 1200 connected online to a Fusion Lumos mass spectrometer (Thermo). Survey scans were collected in data-dependent top-speed mode with dynamic exclusion at 90 s. MaxQuant version 1.6.0.1 search against Mouse Fasta database was used to analyze raw data, with label-free quantification and match between runs functions enabled. DEP package was used to analyze and visualize the output protein group. Processed IP-MS data are provided in Source data file.

## ATAC-seq and analysis

ATAC-seq was performed according to the previous study[42]. Briefly, a total of 50,000 cells were collected and incubated with 50 μL lysis buffer (10 mM Tris-HCl pH 7.4, 10 mM NaCl, 3 mM $MgCl_2$, 0.2% (v/v) IGEPAL CA-630) for 10 min on ice. The suspension of nuclei was centrifuged for 5 min at $500 \times g$ at 4 °C, then resuspended in 50 μL transposition reaction mix (10 μL 5x TTBL, 5 μL TTE Mix V50 and 35 μL nuclease-free $H_2O$) from the TruePrep DNA Library Prep Kit V2 for Illumina (Vazyme, TD501-TD503), and incubated at 37 °C for 30 min. DNA fragments were then PCR amplified and purified with a QIAquick PCR (QIAGEN, 28104) column. Library concentration was detected by a VAHTS Library Quantification Kit for Illumina (Vazyme, NQ101-106) according to the manufacturer's protocol. Finally, libraries were indexed using TruePrep Index Kit V2 for Illumina (Vazyme, TD202), and sequencing was performed on Illumina Novaseq PE150 platform.

The ATAC-seq reads were trimmed by Trim Galore (v0.6.4) and then mapped to the mm10 reference genome using bowtie2 (v2.4.5)[43], and SAMtools (v1.16.1)[44] was used to remove the repetitive, low sequencing quality (mapq < 30) and the mitochondrial DNA mapped reads in the total mapped reads. To make the data comparable between different sequencing depths, the signals were normalized to one million reads for each sample, and the value were further compressed into a binary format (bigWig) for downstream analysis and data visualization. Peak calling was performed using MACS (v1.4.2)[45] with parameters as follows:−g mm−keep-dup all−nomodel−shiftsize 25.

## CUT&Tag and data analysis

CUT&Tag library construction was performed using Hyperactive Universal CUT&Tag Assay Kit for Illumina (Vazyme, TD903) and TruePrep Index Kit V2 for Illumina (Vazyme, TD202)[46]. In brief, around 10000 living cells per sample were obtained and mixed with activated Concanavalin A beads, followed by sequentially incubated with primary and secondary antibodies. The beads were then washed and resuspended in pA(G)-Tn5 buffer supplemented with 10 mM $MgCl_2$ for DNA fragmentation. Next, DNA fragments were indexed using TruePrep Index Kit V2 for Illumina (Vazyme, TD202) and PCR amplified. After purified by VAHTS DNA Clean Beads (Vazyme, N411), libraries were subjected to Illumina Novaseq instruments for sequencing. Antibodies against NANOG (Abcam, ab214549, 1:50), anti-H3K4me1 (Abcam, ab8895, 1:50), anti-H3K27ac (Abcam, ab4729, 1:50), anti-FLAG (Sigma, F1804, 1:50), and anti-SMARCA4 (Cell Signaling Technology, 52251, 1:50) were used in this study.

The CUT&Tag reads were trimmed by Trim Galore (v0.6.4) and then mapped to the mm10 reference genome using bowtie2 (v2.4.5). SAMtools (v1.16.1) was used to remove the repetitive, low sequencing quality (mapq < 30) and the mitochondrial DNA mapped reads in the total mapped reads. The value was further compressed into a binary format for downstream analysis and data visualization. Replicates were merged using samtools merge and then peak calling was performed using MACS (v1.4.2) with parameters as follows:−g mm−keep-dup 1−nomodel−shiftsize 25. The signals were normalized to one million reads for each sample. Promoters were defined as regions ± 2 kb around transcription start sites (TSSs) of genes.

## Motif analysis

Motif scans were performed using HOMER (v4.11.1)[47] against the genome sequence of the given ATAC-seq peaks covered regions (summits ± 25 bp) with the following parameters:−size given−mask. HOMER used hypergeometric test to determine the motif enrichment and test the similarity between the motif we identified to known factors. Motifs which have $p$-value < $10^{-5}$ and enrichment score > 3 are presented in the plot.

## Gene ontology analysis

Functional annotation was performed using the clusterProfiler (v4.6.2)[48]. Gene Ontology terms for each functional cluster were summarized to a representative term, and adjusted $p$-values were plotted to show the significance.

## Cell fate regulator candidates' prediction

Transcription factors whose binding sites are strongly correlated with the specific open loci of terminal cells are identified as candidates for cell fate regulators. For our reprogramming path, we initially selected the ESC-specific open regions compared to the MEF cells. Then we queried the similarity between these regions and the binding regions of factors using the public ChIP-seq database Cistrome. The GIGGLE score (a composite of −log10 $p$-value and log2 odds ratio after querying given regions in the highly confidential peaks of each factor)[49] was used to rank the cell fate regulator candidates.

## Region associated genes

Genes were identified as associated genes with a region when this region suited the pattern proposed by GREAT in the upstream to 5 kb, downstream to 2 kb, and distal to 100 kb of the TSS. Those genes whose promoters were overlapped with the region were also included.

## Enrichment analysis

Enrichment analysis was conducted to evaluate whether the regions in set A overlap with the regions in set B more than expected. Enrichment z-score was calculated using the R library regioneR[42].

## Flow cytometry analysis

Cells were blocked by PBS supplemented with 1% FBS, followed by incubation with antibody against SSEA-1 (Biolegend, 125607, 1:100) for 1 h at 4 °C. After centrifugation, cells were washed twice with PBS, resuspended with PBS containing 0.1% BSA, filtered using a cell strainer (BD Biosciences). The cells were then analyzed using a CytoFLEX 6 L flow cytometer (BD Biosciences). The GFP fluorescence intensity was detected in the FITC channel. The SSEA-1 intensity was detected in the APC channel. Data analysis was performed using FlowJo v.7.6.1.

## Generation of chimeric mice

For chimeras' generation, iPSCs were injected into ICR blastocysts and transplanted into pseudopregnant ICR females. The resulting chimeric mice were determined for color transmission in F1 or germline transmission by mating F1 mice with ICR mice.

## Statistical analysis

Statistical analysis was done using Prism v8.3.0 and R software v4.0.5. Data are represented as mean ± standard deviation. A two-tailed unpaired Student's $t$-test is used to determine the significance between two groups. $P < 0.05$ is considered to indicate a statistically significant value. No statistical method was used to predetermine

sample size. No specific randomization or blinding protocols were used.

## Reporting summary

Further information on research design is available in the Nature Portfolio Reporting Summary linked to this article.

## Data availability

The RNA-Seq, CUT&Tag, ATAC-seq data generated in this study have been deposited in the Gene Expression Omnibus database under the accession number GSE243517. The mass spectrometry proteomics data have been deposited to the ProteomeXchange Consortium via the iProX partner repository with the dataset identifier PXD046008. All other data generated in this study are available in the Supplementary Information or Source data file. Source data are provided with this paper.

## Code availability

The code generated in this study is available on GitHub [https://github.com/LonlyWinter/eCR_Code].

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

## Acknowledgements

We would like to thank all the members of Pei Lab for their helpful discussion and valuable feedback. We thank faculty members at the Biomedical Research Core Facilities and Laboratory Animal Resource Center of Westlake University for assistance. This work was supported in by the National Natural Science Foundation of China (92068201 to D.Q.P.), the Key RD Program of Zhejiang (2024SSYS0029 to D.Q.P.), and Yangtze River Delta Sci-Tech Innovation Community Joint Research Project (2022CSJGG1000 to C.C.Z.).

## Author contributions

T.H., C.C.Z., B.W. and D.Q.P. designed the study. T.H., X.M.W., J.Q.K., Y.X.F., W.H.Z., M.Q.W., B.B.W., B.C., Z.Y.M, Y.F., J.M., Y.Q. and Z.C.L. performed the experiments. T.H., D.L., B.W. and C.C.Z. analyzed data. T.H., C.C.Z., B.W. and D.Q.P. wrote the manuscript. C.C.Z., B.W. and D.Q.P. secured funding and supervised this work.

## Competing interests

The authors declare no competing interests.
