## [Peer Review File · Nature Communications]

REVIEWER COMMENTS

Reviewer #1 (Remarks to the Author):

In this manuscript, the authors reported a novel approach to address the role of the cell intrinsic factors in the reprogramming process. It was known that the BAF complex enhances the reprogramming event mediated by the Yamanaka factors. For example, addition of Brg1 and Baf155 to OSK enhances reprogramming efficiency (Singhai et al, Cell, 2010). The authors' group also reported that SS18 subunit of BAF chromatin remodeling complex regulates reprogramming process (Kuang et al, Nat Commun, 2021). Here they made the fusion protein of Nanog and the BRG1-interacting domain of SS18 (BID) and assessed its ability to promote reprogramming. As the result, they found that the combination of Oct4 and Nanog-BID promotes the reprogramming of MEF to iPS cells as efficient as OSK by recruiting the BAF complex to the wide range of the target sites.

The authors stated that the aim of their research is generating a novel cell-fate regulator that may help us control cell fate rationally. Indeed, Nanog-BID works as a potent reprogramming factor in combination with Oct4 although the wild-type Nanog has low/no potential. However, at the same time, they showed the limitation of this strategy because only Nanog-BID and Oct4-BID works in particular context among the 11 TF-BID fusions tested. The low efficiency of generation of functional regulator is the weakness of this report. In addition, the lack of the validation of the TF-BID fusion genes that fail to enhance the reprogramming event is the flaw of this manuscript. How about their stabilities and DNA binding abilities? Functional assessment of the functions of fusion proteins will be suitable if possible.

This reviewer thinks that further improvement of the efficiency will be required to potentiate their strategy that reach to the level for publication in Nature Communications.

Reviewer #2 (Remarks to the Author):

The authors describe the design and testing of an engineered cell fate controller, NanogBiD, which fuses the BRG1 interacting domain of SS18 with Nanog. This engineered protein efficiently promotes MEF cell reprogramming, in contrast to the native protein, under various testing conditions. The mechanism of action involves recruiting the BAF complex to modulate chromatin accessibility and reorganize cell state-specific enhancers.

The idea of generating the NanogBiD fusion protein is interesting and original literature, however some limitations should be more clearly stated in the manuscript:

- The choice of mouse embryonic fibroblasts (MEFs). Indeed, the inherent plasticity of MEFs, compared to genuine somatic cells, potentially skews the results, leading to conclusions that may not be entirely applicable to other cell types. This discrepancy raises concerns regarding the validity and generalizability of the findings derived from such a model system.

- Moreover, the manuscript states in the introduction the importance of cell reprogramming for health-related applications, hence it is crucial to mention the species-specific differences in reprogramming mechanisms between mice and humans. The cellular machinery and epigenetic landscapes differ significantly between these species, leading to distinct reprogramming dynamics and requirements. As a result, findings obtained from mouse models, such as the one used in this study, may not necessarily translate to human biology.

Reviewer #3 (Remarks to the Author):

The emerging paradigm of cells as drugs necessitates novel strategies to engineer cell-fate in precision. In this manuscript, by fusing Brg1 interacting domain (BiD) of SS18 into the N-terminal of Nanog, the authors designed and validated a new generation of cell-fate controller (NanogBiD) that promote somatic cell reprogramming. The authors further analyzed that eCR facilitate cell fate transition by modulate chromatin accessibility and reorganize cell state specific enhancers.

The design of NanogBiD takes into account the modular nature of protein structures, showcasing a deep understanding of molecular interactions. Moreover, The authors have undertaken a multilayered exploration of the mechanism of NanogBiD, including molecular interactions, chromatin accessibility regulation, and gene activation. This in-depth mechanistic study not only enhances the persuasiveness of the results but also provides the scientific community with a more comprehensive understanding. In addition, NanogBiD uncovers the activation of novel genes like Sox15 and Foxb1, broadening our knowledge of cellular fate control. Finally, the manuscript engages in forward-thinking discussions about the application of engineered chromatin regulators in the era of cells as drugs, which demonstrates the authors' insightful understanding of the practical applications of the research, guiding the future development in the field of translational medicine.

On the whole, the manuscript is innovative, however, there are some issues have to be addressed before publication.

Major points:

1. In the introduction, the authors need further discussion and description as to why CAD serve as a guide to design a new generation of cell-fate regulators. In addition, this part mainly focuses on introducing the aspects from which the new generation of methods can be designed, without comparing with the previous generation of methods to explain whether the previous problems have been solved. A comparative analysis would provide insights into the uniqueness and advantages of NanogBiD.

2. How is the reprogramming strategy determined in Fig 1b? Why a combination of BiD +TFs? How is the number of TFs considered? The author needs more detailed introduction and discussion.

3. I am curious about how the 11 factors are selected in Fig 1c? How do they relate to the 20 factors of Fig S1a? In addition, the title of Fig1 aims to study the importance of CAD. What is the connection between the experimental results of Fig1 and CAD?

4. The article lacks a thorough discussion of potential limitations associated with the use of NanogBiD. Addressing issues such as off-target effects, long-term stability, or scalability would provide a more comprehensive view of the technology's practicality. While the article hints at the potential application of engineered chromatin regulators in the "cells-as-drugs" paradigm, a more explicit discussion on the translational aspects and potential challenges in clinical applications would enhance the article's practical relevance.

5. The Figure Legends in Fig S3 and S4 are particularly confusing, with some legends missing while the figure and legend do not correspond.

Minor points :

1. The definitions of BiD in the text are inconsistent. In Line 55 was represented as "Brg1 interacting domain" while Line 78 was represented as "BAF interacting domain" .

2. In Fig 1d, the culture medium replacement strategy seems to show that the number of clones is the highest when D10 is replaced with N2B27 than D7. Also, would it be more helpful to change the left and right directions of the graph (Days from 0 to 12)? Finally, the chart should be marked with the coordinates "iPSC colonies" in the top right corner.

3. There are no horizontal and vertical coordinates in Fig S2a, besides, it is better to have a statistic.

4. There is lack of reference in line 86-87: "The reprogramming efficiency is approaching to that of OKS which we have shown previously to be superior to the original OKSM, the gold standard combination for iPSC generation."

5. The case of many words in the manuscript is not uniform, such as BAF or Baf; SS18 or Ss18; BRG1 or Brg1.

6. The description of the result of RFX1 in Line 158 159 seems to be inconsistent with the figure, please check.

7. Description of Fig 3d, Fig 3k and Fig S3h is missing in the manuscript.

8. The description of Line174-175 in this manuscript seems to be inconsistent with the result displayed in Fig S3j. Please check.

9. Words that represent genes need to be changed to italics, such as line231.

10. Nanog+Oct4 is referred to as "Nanog", "NanogWT" and " WT " respectively, and it is recommended to unify.

11.The concept of eCR and eCFC is inconsistent. It is defined as “engineered cell-fate regulator or eCR” in the Introduction, while it is "engineered cell fate controller" in the Discussion.

12.In addition to using ESC as a positive control, did the authors consider comparing iPSCs obtained by BiD with iPSCs obtained by OKS? Whether this can better highlight the advantages of eCFC compared with the first generation of cell-fate controller. (such as Fig S1d.)

Dear editor and reviewers,

We would like to thank you for the time and effort you have spent reviewing our paper "Engineering Cell Fate Controller by Rational Design". We appreciate very much your valuable comments, which are extremely helpful for us to improve our manuscript. As suggested by the reviewers, we have conducted additional experiments and analyses to address the questions raised. The manuscript has been revised carefully, and a point-by-point response is provided below. We trust that your comments have been accurately addressed. Please note that the comments are indicated in blue color within the text, while our responses are in black. References are listed at the end of the text, and all figures mentioned have been updated in the new version.

Followings are our point-to-point responses to your comments.

Reviewer #1:

In this manuscript, the authors reported a novel approach to address the role of the cell intrinsic factors in the reprogramming process. It was known that the BAF complex enhances the reprogramming event mediated by the Yamanaka factors. For example, addition of Brg1 and Baf155 to OSK enhances reprogramming efficiency (Singhai et al, Cell, 2010). The authors' group also reported that SS18 subunit of BAF chromatin remodeling complex regulates reprogramming process (Kuang et al, Nat Commun, 2021). Here they made the fusion protein of Nanog and the BRG1-interacting domain of SS18 (BID) and assessed its ability to promote reprogramming. As the result, they found that the combination of Oct4 and Nanog-BID promotes the reprogramming of MEF to iPSC cells as efficient as OSK by recruiting the BAF complex to the wide range of the target sites.

The authors stated that the aim of their research is generating a novel cell-fate regulator that may help us control cell fate rationally. Indeed, Nanog-BID works as a potent reprogramming factor in combination with Oct4 although the wild-type Nanog has low/no potential. However, at the same time, they showed the limitation of this strategy because only Nanog-BID and Oct4-BID works in particular context among the 11 TF-BID fusions tested. The low efficiency of generation of functional regulator is the weakness of this report. In addition, the lack of the validation of the TF-BID fusion genes that fail to enhance the reprogramming event is the flaw of this manuscript. How about their stabilities and DNA binding abilities? Functional assessment of the functions of fusion proteins will be suitable if possible.

Response: We appreciate your insightful comments and suggestion. We agree that only limited number of TFs works when fused to BiD. However, we wish to point out that "the low efficiency" or low yield of functional regulators may be viewed as the following: First, in the context of iPSC generation, the two factor combinations we generated are in fact not very low as factors capable of reprogramming are actually quite rare in nature. For instance, in Yamanaka's original paper, 24 factors were used and 4 became the classic Yamanaka factors (OSKM) ¹. In our panel of 11, two came out, i.e., Oct4^{BiD} and Nanog^{BiD}, a ~20% yield for iPSCs, comparable to that of OSKM out the 24-factor set at 4/24=~ 17%. Secondly, we speculate that the rest can be eCRs for other cell types. If we were to generate neuronal stem cells, Sox2^{BiD} would be a strong candidate². Similarly,

Nr5a2^{BiD} could be good for pancreatic cells³ and Prdm14^{BiD}/ Zfp296^{BiD} for germ cell formation⁴ and spermatogenesis⁵, and so on and so forth. So, we respectfully suggest that the reviewer takes these possibilities into consideration.

We also have performed experiments to address the specificity issue raised by the reviewer. Fusing BiD with TFs tend to remodel the epigenetic microenvironment without changing the TF-intrinsic properties of chromatin occupancy. As shown in figure below, we selected four TFs, including Nanog, Oct4, and two factors that fail to enhance the reprogramming event (*Nr5a2* and *Sox2*), and performed CUT&Tag assay in MEF cells. The *de novo* motif analysis of *Nanog* and Nanog^{BiD} peaks confirmed the enrichment of the NANOG binding motif. Similarly, the *Nr5a2*, Nr5a2^{BiD}, *Sox2*, Sox2^{BiD}, *Oct4*, Oct4^{BiD} peaks showed enrichment of their respective motifs (Fig C1).

CUT&Tag	De novo Motif	Matches to Known Motifs	%T/%B	-log10(pval)
Nanog		NANOG(Homeobox)/mES-NANOG-ChIP-Seq (GSE11724)	1.33	275.0
Nanog ^{BiD}		NANOG(Homeobox)/mES-NANOG-ChIP-Seq (GSE11724)	1.76	>1000
Nr5a2		NR5A2(NR)/mES-NR5A2-ChIP-Seq(GSE19019) NR5A2(NR)/Pancreas-LRH1-ChIP-Seq(GSE34295)	3.12	>1000
Nr5a2 ^{BiD}		NR5A2(NR)/Pancreas-LRH1-ChIP-Seq(GSE34295) NR5A2(NR)/mES-NR5A2-ChIP-Seq(GSE19019) NR5A2/MA0505.1/Jaspar	2.94	>1000
Sox2		SOX2/MA0143.4/Jaspar	2.35	>1000
Sox2 ^{BiD}		SOX2(HMG)/mES-SOX2-ChIP-Seq(GSE11431)	4.21	>1000
Oct4		OCT4(POU,Homeobox)/mES-Oct4-ChIP-Seq (GSE11431)	2.46	>1000
Oct4 ^{BiD}		OCT4(POU,Homeobox)/mES-Oct4-ChIP-Seq (GSE11431)	3.29	>1000

Figure C1. The *de novo* motif enrichment analysis was conducted on CUT&Tag peaks of various factors using HOMER to compare BiD and original TFs, with p-values determined by binomial test. %T/%B represents the detected percentage of targets divided by the detected percentage in the background.

We also addressed stability and functional activities by experiments as shown in figure C2. Fusion with BiD did not affect the expression of such TFs at protein level. Then luciferase reporter assay was performed to detect whether BiD fusion would affect TF binding to its target at the promoter. Methodologically, two plasmids were transfected into 293T cells and then cell lysates were used to measure the luciferase activity based on the amount of bioluminescent signal. One plasmid expresses the WT or BiD fusing TF that would bind to specific promoter sequence. Another plasmid contains promoter sequence fused with the DNA coding sequence for a luciferase enzyme. The bioluminescent signal directly corresponds with the effect of the protein in the binding to promoter. When the 293T cells were transfected with *Oct4*, luciferase activity of *Nanog* promoter was increased about 2 folds as compared to the control ($p < 0.01$). Besides, compared to the Oct4 group, a 50% increase was observed in the cells with Oct4^{BiD} transfection ($p < 0.001$). Luciferase activity of *Sulf* promoter was also significantly increased in 293T cells after transfected with *Oct4* plasmid, whereas a higher elevation of luciferase activity was detected in cell with Oct4^{BiD} transfection. Similar results were observed when we compare the transcriptional activity of other three TFs,

suggesting that BiD fusion did not alter the binding much.

Figure C2. The effect of BiD fusion on the expression and function of original TFs. Western blot analysis showed the change of Oct4 (a), Nanog (b), Sox2 (c) and Nr5a2 (d) protein expression in 293T cells, respectively, under condition of wildtype or BiD overexpression for 3 days. Dual-luciferase reporter analysis of Oct4 (a), Nanog (b), Sox2 (c) and Nr5a2 (d) or their BiD fusion types. 293T cells overexpressing original or BiD expression plasmid or control vector were transfected with the pGL3-promoter of TF target genes. 48 h after transfection, cells were subjected to dual luciferase analysis. Results were expressed as a fold induction relative to the cells transfected with the control vector after normalization to Renilla activity. The experiments were performed independently three times.

Reviewer #2:

The authors describe the design and testing of an engineered cell fate controller, NanogBiD, which fuses the BRG1 interacting domain of SS18 with Nanog. This engineered protein efficiently promotes MEF cell reprogramming, in contrast to the native protein, under various testing conditions. The mechanism of action involves recruiting the BAF complex to modulate chromatin accessibility and reorganize cell state-specific enhancers. The idea of generating the NanogBiD fusion protein is interesting and original literature, however some limitations should be more clearly stated in the manuscript:

- The choice of mouse embryonic fibroblasts (MEFs). Indeed, the inherent plasticity of MEFs, compared to genuine somatic cells, potentially skews the results, leading to conclusions that may not be entirely applicable to other cell types. This discrepancy raises concerns regarding the validity and generalizability of the findings derived from such a model system.

Response: Thank you for your valuable advice about the applicability of other cell types. Accordingly, we isolated mouse tail tip fibroblasts (TTFs) and mouse neonatal fibroblasts (MNFs) from OG2 transgene mice, to investigate the potential of our system in reprogramming other cell types. Our results demonstrate that all these types of cells could be efficiently reprogrammed to Oct4-GFP positive iPSC colonies that can be stably passaged with key pluripotency features (Fig S1g, S1h, S1i), indicating the validity and generalizability of our system.

Supplementary figure 1. Characterization of BiD factors induced iPSCs. **g.** Number of iPSC colonies induced from MNFs and TTFs infected by Nanog^{BiD}+Oct4 or OKS on day 12. *p < 0.05. NS, nonsignificant. N = 3. **h.** Represented images of iPSCs from MNFs and TTFs. Scale bar, 250 μ m. **i.** PCA analysis of RNA-seq data of mESC, and iPSCs induced from MNFs and TTFs infected by Nanog^{BiD}+Oct4 or OKS.

- Moreover, the manuscript states in the introduction the importance of cell reprogramming for health-related applications, hence it is crucial to mention the species-specific differences in reprogramming mechanisms between mice and humans. The cellular machinery and epigenetic landscapes differ significantly between these species, leading to distinct reprogramming dynamics and requirements. As a result, findings obtained from mouse models, such as the one used in this study, may not necessarily translate to human biology.

Response: We appreciate your suggestion and acknowledge that there are still additional barriers to translating this study into human biology. In our research, we found that the current methods and combination of BiD factors effectively reprogram murine cells. As suggested, we have provided forward-thinking discussions on the application of our tool as well as the limitations in the discussion section.

Reviewer #3:

The emerging paradigm of cells as drugs necessitates novel strategies to engineer cell-fate in precision. In this manuscript, by fusing Brg1 interacting domain (BiD) of SS18 into the N-terminal of Nanog, the authors designed and validated a new generation of cell-fate controller (NanogBiD) that promote somatic cell reprogramming. The authors further analyzed that eCR facilitate cell fate transition by modulate chromatin accessibility and reorganize cell state specific enhancers. The design of NanogBiD takes into account the modular nature of protein structures, showcasing a deep understanding of molecular interactions. Moreover, the authors have undertaken a multilayered exploration of the mechanism of NanogBiD, including molecular interactions, chromatin accessibility regulation, and gene activation. This in-depth mechanistic study not only enhances the persuasiveness of the results but also provides the scientific community with a more comprehensive

understanding. In addition, NanogBiD uncovers the activation of novel genes like Sox15 and Foxb1, broadening our knowledge of cellular fate control. Finally, the manuscript engages in forward-thinking discussions about the application of engineered chromatin regulators in the era of cells as drugs, which demonstrates the authors' insightful understanding of the practical applications of the research, guiding the future development in the field of translational medicine. On the whole, the manuscript is innovative, however, there are some issues have to be addressed before publication.

Major points:

1. In the introduction, the authors need further discussion and description as to why CAD serve as a guide to design a new generation of cell-fate regulators. In addition, this part mainly focuses on introducing the aspects from which the new generation of methods can be designed, without comparing with the previous generation of methods to explain whether the previous problems have been solved. A comparative analysis would provide insights into the uniqueness and advantages of NanogBiD.

Response: We thank the reviewer for the suggestion and added the comparative discussion in the revised introduction.

2. How is the reprogramming strategy determined in Fig 1b? Why a combination of BiD +TFs? How is the number of TFs considered? The author needs more detailed introduction and discussion.

Response: In this study, we initially attempted cell reprogramming using a single BiD factor, but only a limited number of clones were produced by Oct4^{BiD}. Therefore, we hypothesized that introducing additional reprogramming factors alongside the BiD factors that may facilitate reprogramming. Subsequently, we tested these BiD factors in combination with 11 wild type TFs (Fig 1c) to convert MEFs to iPSCs and demonstrated that Oct4^{BiD} with *DsRed*, *Esrrb*, or *Nanog*, as well as Nanog^{BiD} with *Jdp2* or *Oct4*, could generate iPSCs with varying efficiency (Fig 1c). As the most efficient combination merged from these attempts is the combination of Nanog^{BiD} *Oct4*, followed by Oct4^{BiD} *Esrrb* (Fig 1c), we decided to focus on the front runner - Nanog^{BiD} *Oct4*. This section has been added to the revised manuscript.

3. I am curious about how the 11 factors are selected in Fig 1c? How do they relate to the 20 factors of Fig S1a? In addition, the title of Fig1 aims to study the importance of CAD. What is the connection between the experimental results of Fig1 and CAD?

Response: Thanks for your suggestion. Based on the concept of CAD, we compared the differences in CAD between mESCs and MEFs to identify chromatin regions that may require opening during cell reprogramming. We then selected candidate factors by referring to binding sites similar to those in the target cell (Fig S1a). In addition to chromatin remodeling factors like *Ss18*, *Smarca4*, *Mbd3*, *Chd7* and *Rnf2*, we found that pluripotent factors *Nanog*, *Sox2*, and *Pou5f1* (*Oct4*) are critical in maintaining open chromatin accessible for transcription in ESC with high GIGGLE score. Considering their importance as core factors for pluripotent maintenance and cell reprogramming, we explored whether fusion the BRG1-interacting domain (BiD) of SS18, a candidate component of CiM previously identified by CRISPR-based screening in a pluripotent-somatic transition system, with these pluripotent factors could achieve cell reprogramming using the GGSGG linker (Fig 1a). Besides, many reported TFs that could facilitate iPSC induction (*Klf4*, *Esrrb*, *Prdm14*, *Zfp296*) or replace *Oct4* (*Nr5a2*, *Jdp2*) were selected for screening. In addition, *Rax* and *Klf17*, which we found they can enhance somatic cell reprogramming (not published results) were also selected. We initially

attempted cell reprogramming using a single BiD factor, but only a limited number of clones were produced by Oct4^{BiD}. Additionally, based on your advice, we have carefully modified the figure legends of our manuscript.

4. The article lacks a thorough discussion of potential limitations associated with the use of NanogBiD. Addressing issues such as off-target effects, long-term stability, or scalability would provide a more comprehensive view of the technology's practicality. While the article hints at the potential application of engineered chromatin regulators in the "cells-as-drugs" paradigm, a more explicit discussion on the translational aspects and potential challenges in clinical applications would enhance the article's practical relevance.

Response: Thank you for your valuable feedback. Accordingly, the discussion section now incorporates additional details regarding the limitations.

5. The Figure Legends in Fig S3 and S4 are particularly confusing, with some legends missing while the figure and legend do not correspond.

Response: Thank you for your valuable feedback. We have carefully modified the figure legends in our manuscript and made the necessary corrections in the revised version.

Minor points:

1. The definitions of BiD in the text are inconsistent. In Line 55 was represented as "Brg1 interacting domain" while Line 78 was represented as "BAF interacting domain".

Response: Thank you for your feedback. We have carefully considered it and made the necessary corrections.

2. In Fig 1d, the culture medium replacement strategy seems to show that the number of clones is the highest when D10 is replaced with N2B27 than D7. Also, would it be more helpful to change the left and right directions of the graph (Days from 0 to 12)? Finally, the chart should be marked with the coordinates "iPSC colonies" in the top right corner.

Response: The number of clones is highest when N2B27 replaces D10, but there is no significant difference between D10 and D7. Additionally, we revised figure 1d based on your suggestion.

3. There are no horizontal and vertical coordinates in Fig S2a, besides, it is better to have a statistic.

Response: Thank you for your advice. We have made careful modifications to the figure and included statistical analysis

4. There is lack of reference in line 86-87: "The reprogramming efficiency is approaching to that of OKS which we have shown previously to be superior to the original OKSM, the gold standard combination for iPSC generation."

Response: The relevant references have been updated and added as per your suggestion. Thank you for your review.

5. The case of many words in the manuscript is not uniform, such as BAF or Baf; SS18 or Ss18; BRG1 or Brg1.

Response: In this paper, we follow this principle: Gene symbols are indicated by italic letters with

the first letter capitalized, but the BiD factors are exhibited by regular letters with the first letter capitalized. Proteins are named with the same symbol as genes, but in full capital letters. The manuscripts have been rechecked and the edit mistakes have been modified, based on your suggestions. Thank you!

6.The description of the result of RFX1 in Line 158 159 seems to be inconsistent with the figure, please check.

Response: Thanks for your advice. We have modified this sentence in our new vision.

7.Description of Fig 3d, Fig 3k and Fig S3h is missing in the manuscript.

Response: The results part of our manuscript has been carefully modified based on your valuable feedback. The necessary corrections have been made in the revised version.

8.The description of Line174-175 in this manuscript seems to be inconsistent with the result displayed in Fig S3j. Please check.

Response: Thank you for your feedback. We have carefully considered it and made the necessary corrections. Thank you!

9.Words that represent genes need to be changed to italics, such as line231.

Response: We reviewed our manuscripts based on your suggestions and corrected these typos. Thank you!

10.Nanog+Oct4 is referred to as "Nanog", "NanogWT" and " WT " respectively, and it is recommended to unify.

Response: According to your suggestions, we have rechecked and modified our manuscripts, making the necessary corrections in the revised version.

11.The concept of eCR and eCFC is inconsistent. It is defined as “engineered cell-fate regulator or eCR” in the Introduction, while it is "engineered cell fate controller" in the Discussion.

Response: Thank you for your feedback. We have carefully considered it and made the necessary corrections. Thank you!

12.In addition to using ESC as a positive control, did the authors consider comparing iPSCs obtained by BiD with iPSCs obtained by OKS? Whether this can better highlight the advantages of eCFC compared with the first generation of cell-fate controller. (such as Fig S1d.)

Response: We appreciate your suggestion and agree that iPSCs obtained by OKS should also be used as a positive control. To address this, we have collected RNA-seq data from three additional cell reprogramming strategies conducted by other research groups: seven factors-derived reprogramming (GSE127927), OKS derived reprogramming (GSE93029), and small-molecule induced reprogramming (GSE48252). As shown in figure S1e, our findings demonstrate that BiD iPSCs exhibit transcriptional similarities to OKS iPSCs and other types of iPSCs, particularly the small-molecule induced iPSCs.

Reference:

- 1 Takahashi, K. & Yamanaka, S. Induction of pluripotent stem cells from mouse embryonic and adult fibroblast cultures by defined factors. *Cell* **126**, 663–676 (2006). [https://doi.org:DOI 10.1016/j.cell.2006.07.024](https://doi.org/DOI%2010.1016/j.cell.2006.07.024)
- 2 Ferrari, F. *et al.* DOT1L-mediated murine neuronal differentiation associates with H3K79me2 accumulation and preserves SOX2-enhancer accessibility. *Nat Commun* **11** (2020). [https://doi.org:ARTN 520010.1038/s41467-020-19001-7](https://doi.org:ARTN%20520010.1038/s41467-020-19001-7)
- 3 Cobo, I. *et al.* Transcriptional regulation by NR5A2 links differentiation and inflammation in the pancreas. *Nature* **554**, 533+ (2018). <https://doi.org:10.1038/nature25751>
- 4 Sybirna, A. *et al.* A critical role of PRDM14 in human primordial germ cell fate revealed by inducible degrens. *Nat Commun* **11** (2020). [https://doi.org:ARTN 128210.1038/s41467-020-15042-0](https://doi.org:ARTN%20128210.1038/s41467-020-15042-0)
- 5 Hackett, J. A. *et al.* Tracing the transitions from pluripotency to germ cell fate with CRISPR screening (vol 9, 4292, 2018). *Nat Commun* **9** (2018). [https://doi.org:ARTN 532810.1038/s41467-018-07765-y](https://doi.org:ARTN%20532810.1038/s41467-018-07765-y)

REVIEWERS' COMMENTS

Reviewer #1 (Remarks to the Author):

In this revised manuscript, the authors tried to answer the questions by the reviewers. However, the answers to the questions of this reviewer sounds superficial and the improvement of the quality of the revised manuscript is quite limited. The present manuscript will be suitable for the publication in Scientific Reports.

Reviewer #2 (Remarks to the Author):

My initial concern was addressed by the authors by conducting additional experiments demonstrating that the approach can also reprogram mouse somatic cells. This response has satisfactorily addressed one of my key concerns.

However, another significant issue remains regarding the relevance of your findings to human biology. I appreciate that the authors have acknowledged this limitation in a new paragraph within the discussion section of the revised manuscript. Nonetheless, I believe that this limitation should be more prominently highlighted to ensure clear communication to readers from the outset. Therefore, I recommend that this limitation also be explicitly mentioned in the abstract.

Furthermore, to prevent any potential misunderstanding about the scope of the study, I suggest revising the title to include the term "mouse." This modification will clarify that the work pertains specifically to mouse cell reprogramming. The current title may be perceived as misleading because it suggests broader applicability without clear indications that the findings are limited to mouse models.

Reviewer #3 (Remarks to the Author):

All my concerns were addressed. I think the paper is ready to be published.

Dear editor and reviewers,

We would like to thank you for the time and effort you have spent reviewing our paper "Engineering Cell Fate Controller by Rational Design". We appreciate very much your valuable comments, which are extremely helpful for us to improve our manuscript. As suggested by the reviewers, the manuscript has been revised carefully, and a point-by-point response is provided below. We trust that your comments have been accurately addressed. Please note that the comments are indicated in black color within the text, while our responses are in blue.

Followings are our point-to-point responses to your comments.

Reviewer #2 (Remarks to the Author):

My initial concern was addressed by the authors by conducting additional experiments demonstrating that the approach can also reprogram mouse somatic cells. This response has satisfactorily addressed one of my key concerns.

However, another significant issue remains regarding the relevance of your findings to human biology. I appreciate that the authors have acknowledged this limitation in a new paragraph within the discussion section of the revised manuscript. Nonetheless, I believe that this limitation should be more prominently highlighted to ensure clear communication to readers from the outset. Therefore, I recommend that this limitation also be explicitly mentioned in the abstract.

Response: We sincerely appreciate your efforts for considering and reviewing our work again. Accordingly, we have checked and modified the abstract in the final version.

Furthermore, to prevent any potential misunderstanding about the scope of the study, I suggest revising the title to include the term "mouse." This modification will clarify that the work pertains specifically to mouse cell reprogramming. The current title may be perceived as misleading because it suggests broader applicability without clear indications that the findings are limited to mouse models.

Response: We sincerely appreciate your advice. Accordingly, we have modified the title as "Engineering Mouse Cell Fate Controller by Rational Design" in the final version.

Reviewer #3 (Remarks to the Author):

All my concerns were addressed. I think the paper is ready to be published.

Response: Thank you for your efforts dedicated to reviewing our manuscript. We are so happy that our previous revised manuscript has provided answers to most of your comments.